# The Effects of Vermicompost and Green Manure Use on Yield and Economic Factors in Broccoli

**Fulya Gul Tascı and Canan Oztokat Kuzucu \***

Department of Horticulture, Agriculture Faculty, Çanakkale Onsekiz Mart University, Çanakkale 17100, Turkey
* Correspondence: coztokat@comu.edu.tr

**Abstract:** The objective of this paper is to evaluate the yield, quality and unit production cost of broccoli grown under green manure, vermicompost and chemical fertilization. For this purpose, broccoli plants were cultivated as follows: (i) Chemical fertilization (CF), (ii) Vermicompost fertilization (V), (iii) Vermicompost fertilization (VeV) in the common vetch cultivated land as a pre-plant and (iv) Vetch (Ve) as a pre-plant. Curd yield (g plant$^{-1}$), SPAD, vitamin C (mg 100 g$^{-1}$), total phenolics (mg GAE 100 mL$^{-1}$), total sugar (%) and soluble solid content (SSC%) were determined. In addition, input usage levels and costs, unit product cost and gross and net profit analyzes were investigated for all treatments. The highest total curd weight was obtained from VeV treatment (1567.77 g), while the lowest mean was reached in Ve plots with 819.70 g. The highest SPAD values were detected in CF (70.20) and VeV plots (69.76) similar to yield values. The highest vitamin C and total phenolics content were detected in V plots (92.31 mg 100 g$^{-1}$) (1308.87 mg GAE 100 mL$^{-1}$). The combined effect of Ve and V increased the yield and quality of the broccoli curds. As a result of economic analysis, approximately twice the net profit difference emerged in VeV treatment compared to other treatments. It was determined that a gross profit of 68% of gross production value and net profit of 64% were obtained.

**Keywords:** *Brassica oleracea* var. *italica*; green manure; vermicompost; profitability; quality

## 1. Introduction

Broccoli (*Brassica oleracea* var. *italica*) is one of the important cool season vegetables of the *Brassicacae* family [1]. Like other *Cruciferous* vegetables, it possesses antioxidant and anti-carcinogenic properties because of its vitamins, carotenoids and polyphenols contents [2–4]. Broccoli consumption and production is increasing day by day in Turkey. Currently, broccoli production is about 120,000 tons in Turkey and consumed mostly as a soup or salad [5]. In addition to the consumer trends, growing cruciferous vegetables is an alternative for farmers for both production and gaining income, especially on the Mediterranean side of Anatolia. Broccoli can be defined as a high nitrogen demanding crop [6] and is greatly sensitive to nitrogen fertilization [7]. Optimum fertilization is necessary for plant growth and development, yield and quality for Brassicas [7]. However, farmers generally prefer to use nitrogen fertilizers uncontrollably in order to increase yield in broccoli production [6,7]. The excessive use of nitrogen causes environmental problems such as ground and surface water pollution, soil salinization and eutrophication because of the high mobility of nitrogen [8,9]. The most striking example of an excessive use of nitrogen is the reported in the Foggia region of Italy, where broccoli is so commonly cultivated that 83.000 ha of land has been identified as sensitive to nitrate [7]. To overcome this problem, it would be beneficial to use plant nutrition alternatives that are, relatively speaking, less harmful to the environment. One of the alternatives is green manure, especially nitrogen (N)-fixing legumes [10]. Green manure crops play an important role for both conventional and sustainable farming systems, and are one of the key elements for ensuring and enhancing sustainability in properly designed crop rotation systems. Many researchers

have reported the benefits of green manure on the physical and chemical properties of the soil [11–14]. The positive effects of green manure on yield, growth, and development are reported in broccoli [15,16]. Moreover, the combined effects of organic matter amendments such as manure, compost and chemical fertilizers with green manure were also reported to be more effective than the fertilizers applied individually [17]. One of the other alternatives to chemical fertilizers is organic fertilization. It is a practical alternative that can help to increase soil productivity and prevent the deterioration of the soil environment. In addition to being a source of nutrients for plants, organic fertilizers provide the development of the physical and chemical properties of the soil [18]. Vermicompost is one of the most important organic fertilizers that have positive effects on the yield and quality traits of vegetables [19–21]. Various results have been reported that vermicompost is individually superior to the farmyard manure on yield and quality in broccoli [22]. As a result, both green manure and vermicompost increase the productivity of broccoli in the case of yield, growth and development. The main reason for increasing yield is related to the amount of organic matter in soil. In this sense, one of the best practices to protect and increase organic matter in the soil is green manuring and incorporating the vermicompost into the soil [23].

The goal of this paper is to determine the effects of vermicompost, vetch as a pre-plant and the combination of them on the yield, quality and net profit in broccoli to overcome the residual effects of excess use of nitrogen fertilizers.

## 2. Materials and Methods

### 2.1. Site Description and Experimental Design

A field trial was conducted in research fields of the Faculty of Agriculture at Canakkale Onsekiz Mart University, Canakkale, Turkey. The fields where the study was carried out are located in the south-western part of Anatolia and in the north-western part of Thrace, at the southern end of the Sea of Marmara ($40°04'24''$ N, $26°21'49''$ E). The study area has a transition climate between the Mediterranean and the Black Sea. The research site is in the Marmara region, which has the 3rd highest vegetable production [24]. Additionally, winter crops generally transplanted into the soil in mid-August in the region and the climatic data, especially the temperature, are very suitable for producing Brassica family vegetables. Four different nutrient treatments were applied to the plots: (i) chemical fertilization (CF), (ii) vermicompost fertilization (V), (iii) vermicompost fertilization (VeV) in the vetch cultivated land as a pre-plant, and (iv) vetch (Ve) as a pre-plant. Before transplanting, half of the nitrogen and full doses of phosphorous and potassium were applied in the plots in CF plots. Nitrogen, phosphorous and potassium were applied in the form of ammonium sulfate, $P_2O_5$ and K+ in CF plots (Gubretas Company, İstanbul Turkey). For Ve plots, vetch (*Vicia sativa* L. cv. Nilufer) seeds were sown at 1.6 kg ha$^{-1}$ for green manuring plots (VeV, Ve) in November 2016 and were plowed into the soil in May 2017, and no amendments were applied into the soil. Vermicompost (Ilpasol; *Eisenia foetida*) was applied to V and VeV plots in doses of 30 kg ha$^{-1}$ one month before the transplanting of seedlings. Vermicompost was applied to V and Ve plots one month before the transplanting of seedlings, as per the manufacturer's recommendation on the product for the decomposition process. The content of the vermicompost is given in Table 1. After the preparation of the plots, broccoli seedlings were transplanted on 22 August 2017 with a 0.33 m intra-row distance and a 0.90 m inter-row distance. Cultivar 'Maraton $F_1$' (Sakata Seeds) was used as a plant material. The main curds were harvested on 2nd December, and lateral curds were harvested on 18 December 2017. The treatments had 3 replicates, randomly distributed in blocks with 20 plants in each repetition, plus side plants. Plants were irrigated with a trickle irrigation system. The plots were kept free of weeds by hand-weeding. The monthly mean air temperature was 17.6 °C, the accumulated rainfall value was 64.72 mm during the cropping season and the total irrigation was approximately 68 dm$^3$ per plant.

**Table 1.** Agrochemical composition of vermicomposused in experiments.

| Parameters | Results |
|---|---|
| pH | 6.80 |
| EC (dS m$^{-1}$) | 5.50 |
| Organic matter (%) | 69.20 |
| Moisture (%) | 66.90 |
| Humic Acid (%) | 13.90 |
| Fulvic Acid (%) | 13.30 |
| Total Nitrogen (%) | 2.20 |
| Total P$_2$O$_5$ (%) | 1.60 |
| Soluble K$_2$O (%) | 1.10 |

### 2.2. Study Parameters

Soil samples were taken and analysed during the cropping season. The first sample was studied before the trials on 2nd August 2017 from the soil depth of 30 cm. Then, the fertilization programme was planned according to the results of this analysis. After green manuring and fertilizing, soil samples were studied and the results are given in Table 2. Total curd weight (g) was determined from 10 plants sampled from the plots and weighted (Vibra RS232C, Shinko, Japan), and total sugar (%), ascorbic acid (mg 100 g$^{-1}$), total phenolics (mg GAE 100 mL$^{-1}$), soluble solid content (SSC %) (Hanna HI 96801), chlorophyll (SPAD) and economic income were evaluated.

**Table 2.** Soil basic analysis at initial and after treatments.

| Treatments | Saturation (%) | EC (mS cm$^{-1}$) | pH | Lime (%) | Organic Matter (%) | P (kg ha$^{-1}$) | K (kg ha$^{-1}$) |
|---|---|---|---|---|---|---|---|
| Initial | 58.00 | 0.64 | 7.58 | 11.80 | 0.96 | 0.74 | 10.92 |
| CF | 59.40 | 0.88 | 7.71 | 12.10 | 0.95 | 0.82 | 13.54 |
| Ve | 63.80 | 1.01 | 7.71 | 11.68 | 2.46 | 1.10 | 13.51 |
| V | 62.10 | 0.98 | 7.83 | 11.74 | 2.87 | 1.10 | 14.20 |
| VeV | 61.23 | 1.23 | 7.79 | 11.79 | 3.35 | 1.62 | 14.89 |

Total sugar was determined by the spectophotometric method explained by Ross (1959) [25], and ascorbic acid was determined by the spectrophotometric method described by Pearson (1970) [26]. Total phenolics were determined by the spectrophotometric method explained by Zheng and Wang (2001) [27] (UV-VIS Spectrophotometer, Shimadzu Corporation, Tokyo, Japan). Spad values were determined by using Konica Minolta SPAD-502 (Konica Minolta, Inc., Osaka, Japan).

### 2.3. Statistical Analysis

Data obtained were processed by SPSS 18.0 (Spss Inc., Chicago, IL, USA) and the variation was classified by Duncan test. Results were evaluated according to $p \leq 0.05$ and $p \leq 0.01$ significance levels.

### 2.4. Economical Evaluation

In order to set an example for farms that grow broccoli with high added monetary value, input usage levels and costs, unit product costs and gross and net profit analyses were investigated [28]. Cost elements, physical input usage levels and unit product costs in broccoli production were calculated, and 2017 data were used in the calculation. Calculations were made by taking into account the labour expended for the production activity of broccoli, machine pulling power, input usage levels, product and input prices and production quantities, in the calculation of the unit cost of broccoli. Family labour for wages and machine power expenses were priced as if the related works were carried out in return for rent, based on the opportunity cost principle. The interest on variable costs

(revolving fund interest) represents the opportunity costs. It expresses the interest income that can be obtained if the amount of production input in question is used in another area. While analysing the input usage in the study, the amount of vetch seed, seedling, chemical fertilizer and vermicompost actually used and the prices paid for them were used. The actual costs incurred by the manufacturer are taken into account in the transportation costs. Fixed costs are the costs incurred even if there is no production in the short term, which is not dependent on the amount of production. Fixed costs in the study consist of depreciation of the irrigation equipment and capital interests, land rent and general administrative expenses. Since the correct line method is used in the depreciation calculation, taking into account the 5% real interest rate while calculating the irrigation tool equipment capital interest, it will be equal to half of the average value costs of the tool equipment throughout its economic life, and the interest calculation has been made by considering half of their values. The revolving fund interest is calculated by using the loan interest rate given by Ziraat Bank for crop production. It is assumed that variable costs are spread over the production period, and interest is calculated based on the interest rate over its half value. General administrative expenses are calculated by taking 3% of the production costs [28]. While calculating the unit costs of broccoli, the simple cost calculation method (total production cost/unit area yield) was used. In order to determine the measure of success in broccoli production activity, gross and net profit calculations were made. Gross profit is calculated by taking the difference between production value and variable costs. Net profit, on the other hand, is calculated by taking the production value and production costs difference [29].

## 3. Results and Discussion

### 3.1. Soil Analysis

Green manuring and fertilizing soil samples were studied, and the results are given in Figure 1. According to the results obtained, except for chemical fertilization before and after all three treatments, the amount of soil organic matter increased significantly after the pre-plant and/or fertilizer treatment compared to the initial values. It is an expected result that the amount of organic matter in the soil will increase after the applications. Many researchers report that green manuring with vetch increases soil organic matter [30,31]. On the other hand, composts are important and, of course, some commercially available organic preparations. Compost materials applied to the soil increase the quality and health of the soil in the long run [32]. From this point of view, it can be claimed that the treatments contribute to the amount of soil organic matter and affect the yield.

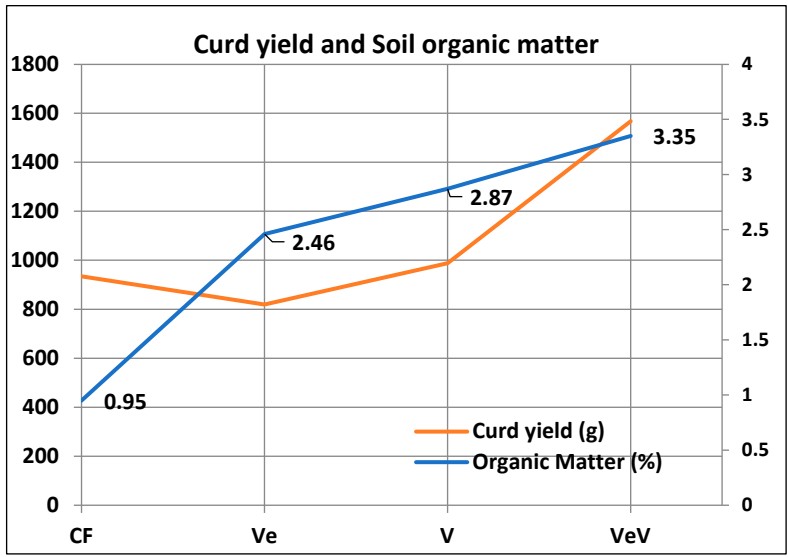

**Figure 1.** Soil organic matter and curd yield changes due to treatments.

### 3.2. Total Curd Yield

Total yield was significantly ($p \leq 0.01$) affected by the treatments (Figure 2). As seen from the data obtained, broccoli grown on vermicompost applied to vetch pre-planted plots (VeV) gave the highest yield per plant (1567.77 g), followed by only vermicompost fertilized plots (V), with the yield of 988.09 g plant$^{-1}$.

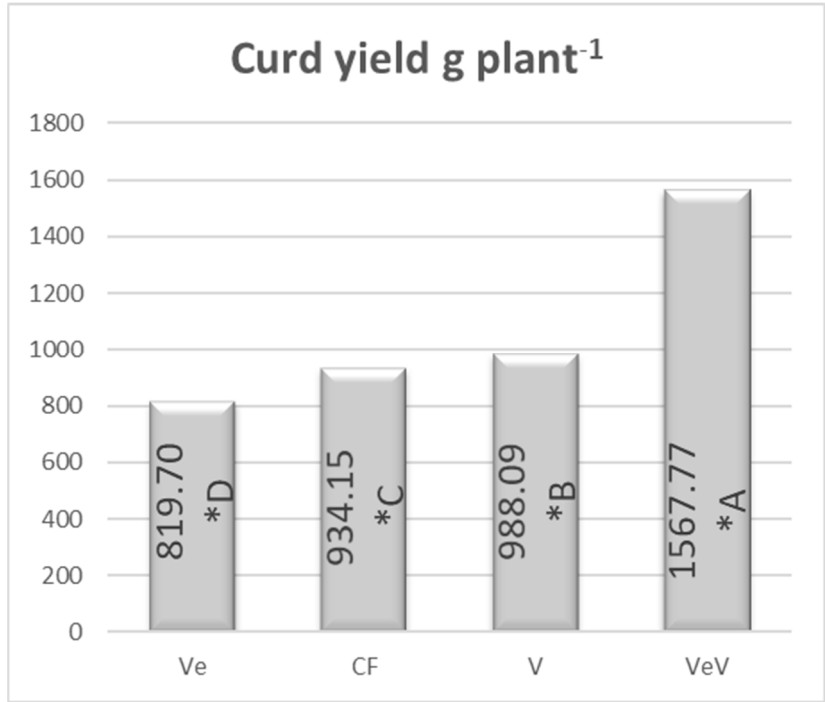

**Figure 2.** Effects of vetch green manuring (Ve), chemical fertilization (CF), vermicompost (V) and vermicompost manuring on vetch pre-planted plots on broccoli yield. * Values followed by different letters in a column were significantly different ($p \leq 0.01$) using Duncan's multiple range test.

The lowest curd weight averages were determined as 934.15 g and 819.70 g, respectively, from the broccoli grown in CF and Ve. When the data obtained were converted to tons per hectare, the yields obtained from the treatments were 46.41 t ha$^{-1}$ for Vev, 27.65 t ha$^{-1}$ for CF, 29.25 t ha$^{-1}$ for V and 24.26 t ha$^{-1}$ for Ve treatments. The curd yield values obtained are partially similar to those in research carried out before [33,34]. On the other hand, broccoli grown in V plots seems to be more productive than the others. When the V effect is combined with vetch, it can be said that the degree of the effect increases. This hypothesis can be claimed because chemical treatments lag behind CF, V and Ve treatments. It would be correct to directly associate the increase in yield with the soil organic matter. Wilkinson (1979) and Arisha et al. (2003) stated that the organic manure's effect on yield may be due to an increase in the organic matter rate in soil [35,36]. In addition, green manuring had positive effects on the yield. However, it seemed that the main effect on yield increase was from vermicompost. Many researchers reported that yield was affected positively by manuring with vermicompost [37,38]. In addition, Zahmancıoglu (2017) investigated the effects of vermicompost and ammonium nitrate applications on growing broccoli and found that the effect of chemical fertilizer treatments on yield was better, whereas vermicompost treatment had positive effects in terms of plant nutrients in the soil and leaves [39]. Additionally, they stated that the highest phosphorus rate in soil was obtained in vermicompost plots. This may explain the yield increase in V and Vev plots in this study. It seemed that the only vetch pre-cropped plots gave the worst results for the curd yield of broccoli. The authors thought that this was a timing problem between the ploughing date of the vetch and the transplanting date of the broccoli seedlings. Sung et al. (2008) incorporated the hairy vetch into the soil as a green manure

and monitored the inorganic nitrogen for 120 days [40]. They reported that the inorganic nitrogen reached the peak around 20 days after incorporation, then decreased after around 30 days after incorporation in soil. This means that the main crop should be transplanted at the most appropriate time after pre crops. In this study, vetch was ploughed into the soil in May 2017 and broccoli seedlings were sown on 22nd August. It seemed that around 120 days is a lot of time for nutrients, especially for inorganic N, to be available for plant absorption. The positive effects of chemical fertilization on curd yield may be explained as the better availability of plant nutrients [36].

The SPAD values of the treatments were given in Figure 3. SPAD values were not significantly affected by the treatments (Figure 3) statistically, but they may be important for explaining the total curd yield. As seen from the data obtained, broccoli grown with chemical fertilization gave the highest SPAD value (70.20), followed by the vermicompost applied on vetch pre-planted plots (VeV) (69.76) and V (69.36). The lowest average was determined as 66.97 from the broccoli grown in V. As noted above, the highest yield was gained from VeV, and the SPAD values determined the second highest. Additionally, VeV, V and CF treatments were very close to each other. The stimulating effect on the chlorophyll content of organic and inorganic fertilizers can be attributed to the fact that N is a component of the chlorophyll molecule, because nitrogen is the main component of all amino acids in proteins and lipids, and acts as a structural component of the chloroplast [14,41]. As explained above, the reason that the lowest yield was obtained in vetch-only fertilization is the missed possibility of the stage where the inorganic nitrogen peaks, according to Sung et al. (2008) [40]. This also seems to have affected the chlorophyll content. On the other hand, CF gave the highest SPAD value; it means that the plant used inorganic N more efficiently than the other treatments, as expected because of nutrient availability. VeV gave the second highest SPAD value, and this result also reflected yield components, but it seems that the Ve effect was limited, and combined with effect of vetch and vermicompost together it increased the SPAD values. This result shows that the VeV application increased soil organic matter more than the others. This resulted in an increase in the cation exchange capacity. It can be said that the mineralization of vermicompost continued, and the inorganic nutrients became more available than the applications of Ve and V alone.

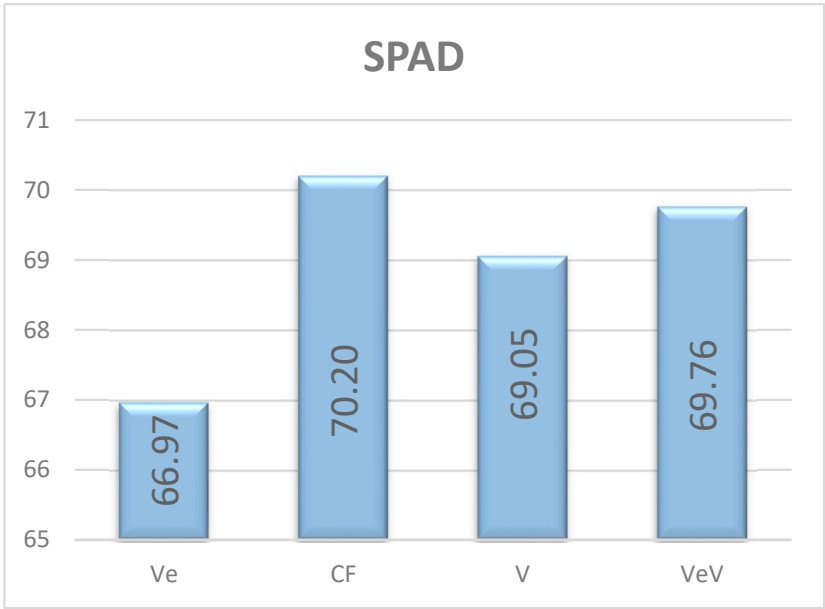

**Figure 3.** Effects of vetch green manuring (Ve), chemical fertilization (CF), vermicompost (V) and vermicompost manuring on vetch pre-planted plots on SPAD values of broccoli.

The effects of vetch green manuring (Ve), chemical fertilization (CF), vermicompost (V) and vermicompost manuring on vetch pre-planted plots on some quality traits of

broccoli are given in Table 3. Quality traits that were examined in this study are vitamin C (mg 100 g$^{-1}$), total phenolics (mg GAE 100 mL$^{-1}$), total sugars (%) and total soluble solid content (SSC %). Vitamin C and total phenolics were affected by the treatment, while the total sugars and SSC were not. The highest Vitamin C value (92.31) was obtained from the vermicompost (V) application, while the lowest was obtained from the vetch (Ve) application.

**Table 3.** Effects of vetch green manuring (Ve), chemical fertilization (CF), vermicompost (V) and vermicompost manuring on vetch pre-planted plots on the curd nutritional quality of broccoli.

| Treatments | Vitamin C (mg 100 g$^{-1}$) | Total Phenolics (mg GAE 100 mL$^{-1}$) | Total Sugar (%) | SSC (%) |
|---|---|---|---|---|
| CF | 80.46 C | 1024.67 B | 2.97 | 7.13 |
| Ve | 80.40 C | 983.23 B | 3.73 | 7.13 |
| V | 92.31 A | 1308.87 A | 3.80 | 7.07 |
| VeV | 90.41 B | 1026.70 B | 3.45 | 6.83 |
| Mean | 85.89 | 1087.87 | 3.49 | 7.04 |
| Degree of Significance | ** | * | NS | NS |

* Values followed by different letters in a column were significantly different ($p \leq 0.05$) using Duncan's multiple range test. ** Values followed by different letters in a column were significantly different ($p \leq 0.01$) using Duncan's multiple range test. NS: not significant.

Vallejo et al. (2003) reported that 'broccoli cv. Marathon' contained 57.40–82.90 mg of vitamin C in 100 g of fresh matter [42]. Our results are parallel to the authors'. There are many results showing that the vitamin C content of vegetables is affected by fertilization. For instance, Mozafar (1993) reviewed and examined the effects of nitrogen fertilization on vitamin C and reported that increased applications of nitrogen fertilizers reduced, increased or had no effect on vitamin C content dependent on the type of plant under study. Although it has been reported in the studies that the amount of vitamin C is affected by nitrogen fertilization, it is known that the effect of light is more efficient during the synthesis of vitamin C by the plant. This situation is reported as a decrease in the synthesis of vitamin C due to the increase in leaf area index as a result of nitrogen fertilization and the decrease in lighting [43]. However, it should be noted that only the amount of vitamin C of the edible part of the plant is measured in the paper. Therefore, in this study, it can be said that the effect of light is less than that of fertilization. The fluctuation seen in vitamin C values can be seen as a result of the organic matter in the soil being in different stages of mineralization.

Total phenolics are presented in Table 3 and expressed as GAE mg 100 g$^{-1}$ FW. Significant differences were found between the treatments. The highest total phenolics content value (1308.87) was obtained from the vermicompost (V) treatment, while the lowest was obtained from the vetch (Ve) treatment. Broccoli sprouts are naturally rich in phenolic compounds, which are susceptible to stress conditions and can be changed with many elicitors [44]. The increase in the TPC can be explained by the role of organic fertilizers in the biosynthesis, which induces the acetate shikimate pathway, resulting in a higher production of flavonoids and phenolics [45].

### 3.3. Economic Analysis

The total production costs in broccoli production activity were analysed as fixed and variable cost elements. While the treatment with the highest production costs was VeV, with 1587.89 USD/da, the lowest was 1102.70 USD/da for the control plot (Table 4). VeV was the highest in the share of variable costs among the production costs, with 89.27%, while the lowest was the CF plot with 85.83%. The treatment with the highest share of fixed costs was CF (14.17%), and the treatment with the lowest ratio was VeV (10.73%). Among the production costs, the highest rate for labour costs were in the parcels of CF and Ve treatments (44.60% and 44.74%, respectively), while material costs were highest in V and VeV treatments (46.74% and 46.55%). The fact that both production costs and

material costs are high in V and VeV treatments is related to the high cost of the applied vermicompost material.

**Table 4.** Production costs per unit area.

| Expense Elements | CF | | Ve | | V | | VeV | |
|---|---|---|---|---|---|---|---|---|
| | Value (USD/ha) | Ratio (%) | Value (USD/ha) | Ratio (%) | Value (USD/ha) | Ratio (%) | Value (USD/ha) | Ratio (%) |
| TOTAL VARIABLE COSTS | 94.67 | 85.83 | 95.49 | 85.92 | 136.64 | 89.00 | 141.75 | 89.27 |
| Labour Costs | 49.19 | 44.60 | 49.73 | 44.74 | 51.35 | 33.45 | 49.19 | 30.98 |
| Machine Pulling Costs | 7.03 | 6.37 | 7.84 | 7.05 | 7.03 | 4.58 | 11.89 | 7.49 |
| Material Costs | 33.95 | 30.78 | 33.38 | 30.03 | 71.76 | 46.74 | 73.92 | 46.55 |
| Revolving Fund Interest | 4.51 | 4.09 | 4.55 | 4.09 | 6.51 | 4.24 | 6.75 | 4.25 |
| TOTAL FIXED COSTS | 15.63 | 14.17 | 15.65 | 14.08 | 16.89 | 11.00 | 17.04 | 10.73 |
| General Administrative Expenses | 2.84 | 2.57 | 2.86 | 2.58 | 4.10 | 2.67 | 4.25 | 2.68 |
| Field Rent | 10.81 | 9.80 | 10.81 | 9.73 | 10.81 | 7.04 | 10.81 | 6.81 |
| Irrigation Tool Machine Ser. Depreciation. | 1.58 | 1.43 | 1.58 | 1.42 | 1.58 | 1.03 | 1.58 | 1.00 |
| Irrigation Tool Machine Ser. Interest | 0.39 | 0.36 | 0.39 | 0.36 | 0.39 | 0.26 | 0.39 | 0.25 |
| TOTAL PRODUCTION COSTS | 110.30 | 100.00 | 111.14 | 100.00 | 153.53 | 100.00 | 158.79 | 100.00 |

Some indicators related to broccoli production activity are given in Table 5. The broccoli yield was highest with VeV application, with 463.91 kg ha$^{-1}$, and lowest in the control application with 276.77 kg ha$^{-1}$. By proportioning the total production costs to the production amount, the cost of one kg of broccoli was found to be the lowest, with 0.34 USD/kg in VeV treatment, and the highest had 0.52 USD/kg in V treatment. Although the production cost of VeV treatment is quite high compared to other treatments, the cost was found to be the lowest due to the extremely high yield. According to the fruit and vegetable market registration data of the Ministry of Commerce of the Republic of Turkey (http://www.hal.gov.tr/ (access on 15 September 2017)), the average selling price of broccoli is about 0.95 USD/kg. It was found that a profit of 0.61 USD/kg was obtained from the VeV treatment. The calculation of gross and net profits per unit area in broccoli growing areas has an important place in determining the income of producers from this production activity. While the gross profit per hectare from the Ve treatment was lowest in broccoli cultivation, at USD 134.26, it was USD 297.09 in VeV. A 121% difference was calculated between the treatments. It was determined that there was a 136% difference in terms of net profitability between the treatments with the highest net profit (280.04 USD/ha) and the treatment with the lowest (118.61 USD/ha).

**Table 5.** Production cost, expenses and revenues per unit area.

| Expense and Income Elements | CF (USD/ha) | Ve (USD/ha) | V (USD/ha) | VeV (USD/ha) |
|---|---|---|---|---|
| Gross Production Value | 261.81 | 229.75 | 276.88 | 438.83 |
| Variable Costs | 94.67 | 95.49 | 136.64 | 141.75 |
| Production Costs | 110.30 | 111.15 | 153.53 | 158.79 |
| Gross profit | 167.14 | 134.26 | 140.24 | 297.08 |
| Net profit | 151.51 | 118.61 | 123.35 | 280.04 |
| Yield (kg/ha) | 276.51 | 242.63 | 292.47 | 464.06 |
| Cost (USD/kg) | 0.40 | 0.46 | 0.52 | 0.34 |
| Relative Profit | 2.37 | 2.06 | 1.80 | 2.76 |

## 4. Conclusions

The treatment with the highest relative profitability was found to be VeV (2.76), and the lowest was the V treatment (1.80). In this case, an income of USD 2.76 is obtained for an expense of USD 1. Production and profit thresholds in these four different treatments in broccoli cultivation are below the average price. In this case, a good profit level was achieved in all treatments. However, in the VeV treatment, approximately two times the net profit difference emerged compared to the others. In the aforementioned treatment, it was determined that 68% of the gross production value and 64% of the net profit were obtained. As a result, it can be seen that the VeV treatment, which gives the highest results in quality and yield parameters and is more profitable in economic evaluation, is more advantageous among the treatments.

In many studies, it has been stated that nitrogen leaching causes environmental pollution. Especially in developed and developing countries, excessive chemical fertilization for crop cultivation becomes even more harmful in plant species with a short growing period, such as broccoli. With the combined effect of organic fertilizers, these harmful effects can be reduced. The most important outcome of this study is that vetch plus vermicompost applications without chemical fertilizers gives better results in terms of both yield and profitability than other applications.

**Author Contributions:** Conceptualization, methodology, supervision, funding acquisition, writing—review and editing performed by C.O.K.; software, formal analysis, investigation, writing—original draft preparation performed by F.G.T. All authors have read and agreed to the published version of the manuscript.

**Funding:** This research received no external funding.

**Data Availability Statement:** The datasets generated for this study are available on request to the corresponding author.

**Acknowledgments:** This study is derived from Fulya Gul Tascı's Master of Science Thesis. The authors wish to thank Duygu Aktürk for the contribution of performing economic analyses.

**Conflicts of Interest:** The authors declare no conflict of interest.

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
