# Peer review of "The Effects of Vermicompost and Green Manure Use on Yield and Economic Factors in Broccoli"

_horticulturae, doi:10.3390/horticulturae9030406_

Round 1

Reviewer 1 Report

73 Vermicompost was treated in plots one month before the transplanting of seedlings. Was vermicompost applied to all the plots? if not you need to indicate plots that were fertilized one month before fertilization and why the practice did not alter plant growth relative to the other treatments.

74-76 According to the brief given above the treatments occurred as follows; i) 

Chemical fertilization (CF), ii) Vermicompost fertilization (V), iii) Vermicompost fertilization (VeV) in the vetch cultivated land as a pre-plant, iv) Vetch (Ve) as a pre-plant... treatment and replication creates tends to suggest broccoli is mostly cultivated using inorganic fertilizer. Is it possible other organic manures are used? what is the difference in performance between manure and vermicompost? you need to explain why you have focused on CF and Vev.

76-77 Before transplanting, half of the nitrogen and full doses of phosphorous and potassium were applied to the plots in CF plots. Could another replication with fertilizer applied after transplanting have given a different outcome? my question is how robust are your conclusions? is the difference between CF and VeV due to the periodization of fertilization or the type of fertilization?

79-80 no any.. should read no amendments....

105 with high added value... are you referring to weight (biomass) or monetary value? a sentence is vague.

 44,74% although it is a matter of style, I still would suggest you use a dot  (44.74% ) in place of a comma as the separator. 

248 there is a need for consistency in the tables. some entries are rounded-up to two decimal places others to three.

255-256 considering that the average selling price of broccoli is about $0,95/kg,.. this claim should be supported by evidence or source.

266-274 first the conclusion is scanty. Secondly, no attempt is made to compare the projections (estimations) with real or market profit levels obtained. 

Author Response

Dear Reviewer;

We author's have tried to make all the corrections you deem necessary and shown below. We hope you find our corrections adequate and appropriate.

See attachment please

Best regards

In this context, this paper discusses the effects of several organic and environmentally friendly fertilizers on broccoli yield, quality, and unit cost, and demonstrates through several controlled experiments that the combination of vetch + vermicompost (VeV) maximizes the economic utility of broccoli. The paper has great potential for publication, and it is recommended to revise it before publication. Specific comments are as follows:

  1. Abstract. The abstract is incomplete and not detailed enough, which should include research background, research methods and content, research objectives, and research results. In particular, the main conclusions of this article are not fully summarized in the abstract. The authors should have added the results of the SPAD and quality traits analysis to the abstract as well。

Abstract rewrited according to the recommendations.

  1. Introduction. The theoretical background and literature review are not comprehensive enough. Relevant literature studies and reviews on the effects of other organic fertilizers and green manure as pre-plant materials on crops should be added appropriately.

Some literatures (9 more) added especially on organic fertilizers. Lines 43-52, 57-61, 65-69

  1. Materials and methods. The conclusions of the article should be further tested. The text uses only data from the 2017 issue for the experiment in calculating cost elements, physical input use levels, and unit product costs in broccoli production, and agricultural cultivation is heavily influenced by the interference of other factors, which makes the robustness of the article's results questionable. The authors should describe why the area was chosen as a test area and whether its climate and soil environment are suitable for growing broccoli.

Calculations related to the economic data in the study were made by considering the Turkish central bank dollar rate during the production period (November 2016-December 2017) in which the research was conducted. Calculations are made in dollars due to the variability in the Turkish lira and exchange rates. While making the calculations, the figures valid in the production year were used.

Descriptions added (82-85) for the test area and climate-soil environment

  1. Conclusion. The discussion and conclusion sections are not sufficiently detailed. There is no discussion of how the results contribute to the existing literature, only the experimental results are reported. Suggestions and insights into the development of this direction should be provided.

This section has been renamed as findings and discussion, since discussion is included in the findings title. Lines 231-239, 261-270 added and also some parts revised. At the same time, an addition has been made to the result section (319-325)

  1. Some Minor comments. (1) Note the neat formatting and typographical issues of the article. The language should be polished, I recommend using Academic English and avoiding personal sentences. For example, Vermicompost fertilization (VeV) in the common vetch cultivated land as a pre-plant should be modified to Vermicompost fertilization(V) in the common vetch cultivated land as a pre-plant (VeV). (2) The format of the graphs in the text should be standardized. Most of the figures in the paper appear to lack legends and data units.

Corrections performed according to the recommendations

Reviewer 2 Report

In this context, this paper discusses the effects of several organic and environmentally friendly fertilizers on broccoli yield, quality, and unit cost, and demonstrates through several controlled experiments that the combination of vetch + vermicompost (VeV) maximizes the economic utility of broccoli. The paper has great potential for publication, and it is recommended to revise it before publication. Specific comments are as follows:

1.     Abstract. The abstract is incomplete and not detailed enough, which should include research background, research methods and content, research objectives, and research results. In particular, the main conclusions of this article are not fully summarized in the abstract. The authors should have added the results of the SPAD and quality traits analysis to the abstract as well。

2.     Introduction. The theoretical background and literature review are not comprehensive enough. Relevant literature studies and reviews on the effects of other organic fertilizers and green manure as pre-plant materials on crops should be added appropriately.

3.     Materials and methods. The conclusions of the article should be further tested. The text uses only data from the 2017 issue for the experiment in calculating cost elements, physical input use levels, and unit product costs in broccoli production, and agricultural cultivation is heavily influenced by the interference of other factors, which makes the robustness of the article's results questionable. The authors should describe why the area was chosen as a test area and whether its climate and soil environment are suitable for growing broccoli.

4.     Conclusion. The discussion and conclusion sections are not sufficiently detailed. There is no discussion of how the results contribute to the existing literature, only the experimental results are reported. Suggestions and insights into the development of this direction should be provided.

5.     Some Minor comments. (1) Note the neat formatting and typographical issues of the article. The language should be polished, I recommend using Academic English and avoiding personal sentences. For example, Vermicompost fertilization (VeV) in the common vetch cultivated land as a pre-plant should be modified to Vermicompost fertilization(V) in the common vetch cultivated land as a pre-plant (VeV). (2) The format of the graphs in the text should be standardized. Most of the figures in the paper appear to lack legends and data units.

Author Response

Dear Reviewer

We tried to make the corrections that is foreseen and shown below. We hope we have performed adequately and correctly. 

Best regards

Response to Reviewers Suggestions

73 Vermicompost was treated in plots one month before the transplanting of seedlings. Was vermicompost applied to all the plots? if not you need to indicate plots that were fertilized one month before fertilization and why the practice did not alter plant growth relative to the other treatments.

Vermicompost only applied to the vermicompost and vetch plots according to the manufacturer’s recommendations for the decomposition.

When vermicompost application was applied alone, it gave higher results than chemical fertilization and only vetch application, however, the plots where vermicompost application was made with vetch showed the highest effect, where it is seen that the application of vermicompost with green manure improves the yield considerably (Lines 197-215)

74-76 According to the brief given above the treatments occurred as follows; i) 

Chemical fertilization (CF), ii) Vermicompost fertilization (V), iii) Vermicompost fertilization (VeV) in the vetch cultivated land as a pre-plant, iv) Vetch (Ve) as a pre-plant... treatment and replication creates tends to suggest broccoli is mostly cultivated using inorganic fertilizer. Is it possible other organic manures are used? what is the difference in performance between manure and vermicompost? you need to explain why you have focused on CF and Vev.

According to the recommendations; “Vermicompost is one of the most important organic fertilizer that have positive effects on yield and quality traits of vegetables [19. 20. 21]. Various results were reported that vermicompost has individually superior than the farmyard manure on yield and quality in broccoli [22]. As a result, both green manure and vermicompost increase the productivity of broccoli in case of yield, growth and development. The main reason for increasing yield is related to to the amount of organic matter in soil. In this sense; one of the best practices to protect and increase the organic matter in the soil is green manuring and incorporating the vermicompost into the soil [23].  ” has added to lines 65-69

76-77 Before transplanting, half of the nitrogen and full doses of phosphorous and potassium were applied to the plots in CF plots. Could another replication with fertilizer applied after transplanting have given a different outcome? my question is how robust are your conclusions? is the difference between CF and VeV due to the periodization of fertilization or the type of fertilization?

The chemical fertilization program was made according to the results of the soil analysis made before the experiment started. Dividing the nitrogen dose into two was made according to the recommendation in the literature, and the fertilization regime applied in the region is also the same in this sense. It is possible that different doses and application times of organic fertilizers may have different effects, and further studies are planned on these issues. Aim of our study is not only compare the effcets of yield but also the profit by using the organic fertilizers in the sense of commercial use.

79-80 no any.. should read no amendments..

Corrected as no amendments..

105 with high added value... are you referring to weight (biomass) or monetary value? a sentence is vague.

Monetary value added to line 129

 44,74% although it is a matter of style, I still would suggest you use a dot  (44.74% ) in place of a comma as the separator. 

All punctuation errors in the text have been corrected.

248 there is a need for consistency in the tables. some entries are rounded-up to two decimal places others to three.

All the decimals are corrected in the tables

255-256 considering that the average selling price of broccoli is about $0,95/kg,.. this claim should be supported by evidence or source.

Sentence added “According to the fruit and vegetable market registration data of the Ministry of Commerce of Republic of Turkey (http://www.hal.gov.tr/) Line 297-298

266-274 first the conclusion is scanty. Secondly, no attempt is made to compare the projections (estimations) with real or market profit levels obtained. 

Arrangements were made in the conclusion section and discussion sections in line with the opinions of the reviewer. LÄ°ne 319-325

.

Reviewer 3 Report

The manuscript titled ‘’ The effects of vermicompost and green manure use on yield and economic factors in broccoli’’ needs some important corrections before further processing.

The abstract needs to be improved. This section is finalized with some results! Nothing about the conclusion.

The introduction needs some serious work. The background of research needs to be strengthened. Some sentences are written in a simple way without scientific soundness. English editing is recommended for this section.  All in all, this section is not even one page. So definitely needs to be extended! The number of references is too low. Moreover, most of them are out of date!

Figures are presented with very poor quality!

Results and Discussion are not well described. The used references are too old!

The conclusion is nothing except repeating the results.

There are some points in the text, please see the attached.

Author Response

Dear Reviewer

We tried to complete all the corrections and suggestions and shown below. Unfortunately we could not perform  your recommendation about SPSS programme that we have only SPSS 18 with a licence. Except this point we tired to do the every correction.

See also attachment

Best Regards

Canan Öztokat Kuzucu

The manuscript titled ‘’ The effects of vermicompost and green manure use on yield and economic factors in broccoli’’ needs some important corrections before further processing.

The abstract needs to be improved. This section is finalized with some results! Nothing about the conclusion.

Abstract rewritten by the recommendations (Lines 8-32)

The introduction needs some serious work. The background of research needs to be strengthened. Some sentences are written in a simple way without scientific soundness. English editing is recommended for this section.  All in all, this section is not even one page. So definitely needs to be extended! The number of references is too low. Moreover, most of them are out of date!

Introduction rewritten and corrected also added new literatures (9 more) Lines; 43-52, 57-61, 65-69

Figures are presented with very poor quality!

Figures are designed again

Results and Discussion are not well described. The used references are too old!

Results and Discussion part revised and some parts added Lines 231-239, 261-270.

The conclusion is nothing except repeating the results.

Conclusion revised and Lines 319-325 added

There are some points in the text, please see the attached.

All the points in the text corrected.

Round 2

Reviewer 3 Report

The manuscript was improved - some general comments from the first review remaining, but the authors improve the manuscript (they also indicated changes in the separate file). I think the manuscript need to be very carefully checked for some spelling errors and other mistakes. For example:

1 Affiliation 1; [email protected] 5

2 Affiliation 2; [email protected]

 [2, 3]. [4]. Broccoli

Author Response

Dear Reviewer;

We carefully check all the text and correct the errors and mistakes. 

Thank you in advance

Best regards

Canan Öztokat Kuzucu
